

# Mapping gaseous amines, ammonia, and their particulate counterparts in marine atmospheres of China's marginal seas: Part 1 - Differentiating marine emission from continental transport

Dihui Chen[1], Yanjie Shen[1], Juntao Wang[1], Yang Gao[1,2], Huiwang Gao[1,2], Xiaohong Yao[1,2*]

[1]Key Laboratory of Marine Environment and Ecology, and Frontiers Science Center for Deep Ocean Multispheres and Earth System, Ministry of Education, Ocean University of China, Qingdao 266100, China
[2]Laboratory for Marine Ecology and Environmental Science, Qingdao National Laboratory for Marine Science and Technology, Qingdao 266237, China

*correspondence to*: Xiaohong Yao (xhyao@ouc.edu.cn)

**Abstract.** To study sea-derived gaseous amines, ammonia, and primary particulate aminium ions in the marine atmospheres of China's marginal seas, an onboard URG-9000D Ambient Ion Monitor-Ion chromatography (AIM-IC, Thermo Fisher) was set up on the front deck of the R/V Dongfanghong 3 to semi-continuously measure the spatiotemporal variations in the concentrations of atmospheric trimethylamine ($TMA_{gas}$), dimethylamine ($DMA_{gas}$), and ammonia ($NH_{3gas}$) along with their particulate matter ($PM_{2.5}$) counterparts. In this study, we differentiated marine emissions of the gas species originating from continental transport using data obtained from December 9 to 22, 2019 during the cruise over the Yellow and Bohai Seas, facilitated by additional measurements collected at a coastal site near the Yellow Sea during summer 2019. The data obtained during the cruise and the coastal site demonstrated that the observed $TMA_{gas}$ and protonated trimethylamine ($TMAH^+$) in $PM_{2.5}$ over the Yellow and Bohai Seas overwhelmingly originated from marine sources. During the cruise, there was no significant correlation (P>0.05) between the simultaneously measured $TMAH^+$ and $TMA_{gas}$ concentrations. Additionally, the concentrations of $TMAH^+$ in the marine atmosphere varied around 0.28±0.18 μg m$^{-3}$ (average ± standard deviation), with several episodic hourly average values exceeding 1 μg m$^{-3}$, which were approximately one order of magnitude larger than those of $TMA_{gas}$ (approximately 0.031±0.009 μg m$^{-3}$). Moreover, there was a significant negative correlation (P<0.01) between the concentrations of $TMAH^+$ and $NH_4^+$ in $PM_{2.5}$ during the cruise. Therefore, the observed $TMAH^+$ in $PM_{2.5}$ was overwhelmingly derived from primary sea-spray aerosols. Using the $TMA_{gas}$ and $TMAH^+$ in $PM_{2.5}$ as tracers for sea-derived basic gases and sea-spray particulate aminium ions, the values of non-sea-derived $DMA_{gas}$ and $NH_{3gas}$,

as well as non-sea-spray particulate $DMAH^+$ in $PM_{2.5}$, were estimated, and the estimated average values of each species contributed to 16%, 34%, and 65% of the observed average concentrations, respectively. Uncertainties remained in the estimations as $TMAH^+$ may decompose into smaller molecules in seawater to varying extents. The non-sea-derived gases and non-sea-spray particulate $DMAH^+$ likely originated from long-range transport from the upwind continents, according to

the recorded offshore winds and increased concentrations of $SO_4^{2-}$ and $NH_4^+$ in $PM_{2.5}$. The lack of a detectable increase in the particulate $DMAH^+$, $NH_4^+$, and $SO_4^{2-}$ concentrations in several $SO_2$ plumes did not support the secondary formation of particulate $DMAH^+$ in the marine atmosphere.

**Keywords:** Marine atmospheric $NH_3$, trimethylamine, dimethylamine, particulate aminium, sea-spray aerosol

## Introduction

Gaseous amines and their particulate counterparts are important reduced nitrogen compounds in the marine atmosphere (Facchini et al., 2008; Müller et al., 2009; Hu et al., 2015; Hu et al., 2018; van Pinxteren et al., 2015; van Pinxteren et al., 2019; Yu et al., 2016; Xie et al., 2018; Zhou et al., 2019) and are primarily derived from the degradation of glycine betaine (GBT), trimethylamine N-oxide (TMAO), and choline (Burg and Ferraris, 2008; Lidbury et al., 2015a; Lidbury et al., 2015b; Jameson et al., 2016; Taubert et al., 2017). GBT, TMAO, and choline are critical for maintaining osmotic pressure in marine

organisms. When released into the environment, they can be degraded by bacteria to trimethylamine (TMA) and then dimethylamine (DMA) or methylamines (MA) (Lidbury et al., 2015a; Lidbury et al., 2015b). Gaseous DMA, TMA, and MA may play an important role in the formation of secondary particles in the atmosphere by nucleation (Almeida et al., 2013; Chen et al., 2016; Yao et al., 2018; Zhu et al., 2019). However, measuring gaseous amines in real-time simultaneously to their particulate counterparts in the marine atmosphere over the ocean remains challenging, although this is not the case in

the continental atmosphere (VandenBoer et al., 2011). The lack of direct measurements restricts the determination of their sources and the relationship between the reduced nitrogen compounds and acid-base neutralization reactions in the marine atmosphere.

Reduced nitrogen compounds in the ocean can finally decompose into ammonium ions ($NH_4^+$) and other smaller molecules.

$NH_4^+$ in surface seawater releases to the marine atmosphere as atmospheric ammonia ($NH_{3gas}$) under favorable conditions

(Johnson et al., 2008; Carpenter et al., 2012; Paulot et al., 2015). The ocean is an important source of $NH_{3gas}$, contributing to

approximately 40% of the natural $NH_3$ emissions on Earth (Carpenter et al., 2012; Paulot et al., 2015). In the literature, large

uncertainties in the estimated $NH_3$ emissions from the ocean remain; for example, the annual emission flux ranges from 2 to

23 Tg N $a^{-1}$; (Clarke and Porter, 1993; Dentener and Crutzen, 1994; Sutton et al., 2013; Paulot et al., 2015). These

uncertainties are primarily derived from two factors: 1) the major marine sources of $NH_{3gas}$ are still disputed, such as

seawater, sea-birds, or the photolysis of marine organic nitrogen at the ocean's surface or in the atmosphere; and 2) direct

observations of $NH_{3gas}$ in marine atmospheres are restricted as onboard ambient $NH_{3gas}$ measurement techniques sometimes

suffer from large artifacts due to $NH_{3gas}$ contamination associated with onboard human activities, dew evaporation, and

interference from water vapor. (Quinn et al., 1990; Clarke and Porter, 1993; Johnson et al., 2008; Keene et al., 2009;

Wentworth et al., 2016; Teng et al., 2017) Additionally, the long-range transport of atmospheric $NH_{3gas}$ from the continent

may also complicate the source analysis of $NH_{3gas}$ in marine atmospheres (McNaughton et al., 2004; Uematsu et al., 2004;

Zhao et al., 2015; Lutsch et al., 2016).

To identify and characterize sea-derived gaseous amines, ammonia and sea-spray particulate aminium ions, as well as

secondary particulate aminium ions from continental transport in the atmospheres of China's marginal seas, we conducted a

cruise campaign over the Yellow and Bohai Seas in China from 9 to 22 December 2019 (Campaign A), and over the Eastern

China and Yellow seas from December 27, 2019, to January 16, 2020 (Campaign B). Winter cruise campaigns provide great

opportunities for observational studies due to the 1) higher concentration levels of nutrients in the seas at a lower sea surface

water temperature (Guo et al., 2020); 2) periodically enhanced air-sea exchange driven by the strong winter Asian monsoon

every 4–10 d (Zhu et al., 2018); and 3) periodically enhanced long-range transport of anthropogenic pollutants from

continents to the seas (Guo et al., 2016; Wang et al., 2019).

In this study, an onboard URG-9000D Ambient Ion Monitor-Ion chromatography (AIM-IC, Thermo Fisher) was used to

simultaneously measure the spatiotemporal variations in the concentrations of gaseous amines and $NH_{3gas}$ with their

counterparts in $PM_{2.5}$. Semi-continuous measurement data were then analyzed to identify the study targets. This study was

divided into two parts. In this part, we focused on identifying marine sources from the continental transport of reduced

nitrogen compounds in marine atmospheres and subsequently quantified each contribution to the observed species during the

9-22 December 2019 campaign. In the companion paper (Gao et al., submitted to ACP), we analyzed the spatiotemporal

heterogeneity and related causes, and then delivered a hypothesis regarding the marine emissions of reduced nitrogen

compounds using the data obtained during the two campaigns and data from an additional cruise campaign previously

reported by Hu et al. (2015).

**Experimental**

Campaign A was conducted from December 9 to 19, 2019, on the R/V Dongfanghong-3 with a displacement tonnage of

5000 (Fig. 1). The research vessel was still within its testing period and used state-of-the-art combustion technology with

low-sulfur diesel. On December 19-22, the vessel was anchored at the port for Campaign B, organized by another research

team. A standard-sized air-conditioned container was set up on the front deck to house a suite of instruments for measuring

the air pollutant concentrations. No human activities occurred on the front deck during cruising, excluding anchoring at the

port. Even during the anchoring period, human activity on the front deck was rare. The use of the container on the front deck

effectively minimized self-vessel contamination by $NH_{3gas}$ and gaseous amines. The front deck was approximately 10 m

above sea-level, and the container height was 2.8 m.

To ensure that the onboard AIM-IC was operating properly, it was housed in a mobile air-conditioned mini-container, which

was further housed in a standard container with a 1-m stainless steel sampling probe connected to the ambient air. The inlet

of the sampling probe extended from the top corner of the standard container facing the sea. A $PM_{2.5}$ cyclone was installed

on the AIM-IC and operated at a rate of 3 L/min, which reports the semi-continuous concentrations of chemically reactive

gases, including $NH_{3gas}$, gaseous amines, and acidic gases such as $SO_2$ and $HNO_3$, along with their particulate counterparts,

at a temporal resolution of 1 h, allowing the identification of possible interference from onboard dew evaporation, which

typically occurs with sunrise (Teng et al., 2017).

The AIM-IC includes an ICS-1100 ion chromatograph, in which an analytical column (Ion Pac CS17A (2×250 mm)) was

used to measure cations, including $NH_4^+$, protonated dimethylamine ($DMAH^+$), and protonated trimethylamine ($TMAH^+$),

and an AG11-HC (2×50 mm) for measuring anions, including $SO_4^{2-}$, $NO_3^-$, $Cl^-$, and organic ions. The detection limits of

$NH_4^+$, $DMAH^+$, and $TMAH^+$ in the injection solution were 0.001, 0.008, and 0.001 mg/L, respectively. The ICS-1100 was

calibrated onboard prior to the commencement of regular measurement collection, and the second calibration was conducted

when the vessel was anchored at the port. The AIM-IC analysis was not affected by ambient water vapor as the device

directly measured the ions. More detailed information regarding AIM-IC analysis is provided in the studies of Teng et al.

(2017) and Xie et al., (2018). It should be noted that strong $K^+$ contamination unexpectedly occurred occasionally and then

disappeared during different campaigns. When contamination occurred, $DMAH^+$ and $TMAH^+$ were undetectable due to the

increased baseline at the corresponding residence time in the ion chromatograph; as such, some $PM_{2.5}$ $DMAH^+$ and $TMAH^+$

concentration data were unavailable in Fig 2. However, the concentrations of gaseous amines were still correctly detected

with a low baseline at the residence. The $K^+$ contamination remains under investigation.

An automatic weather system that provides real-time meteorological data is available on the R/V Dongfanghong-3. The

heading wind was corrected to determine the true wind speed and direction. The surface seawater temperature was not

measured during this cruise campaign, and typically has a delay of a few hours when compared to the ambient air

temperature (Deng et al., 2014). The $NH_4^+$ and aminium ion concentrations in the surface seawater were also not measured

as the analytical methods are still hindered by high sea-salt ion contents.

On August 1-9 and September 12 to October 1, 2019, the AIM-IC was set up at a coastal site in Qingdao (36.34°N, 120.67°E)

to collect routine measurements (Fig 2). The summer measurement data were obtained three to four months before the winter

cruise campaign. The sampling site was located in a new high-technology zone near the Yellow Sea, with the shortest

distance from the sea being approximately 1 km in the south. The AIM-IC was housed in a research lab on the fifth story of a

building, approximately 16 m above ground-level. The sampling probe extended out of the window and was directly

connected to the ambient air. Typically, higher biogenic emissions of reduced nitrogen compounds over the continents are

expected in the summer than the winter due to the temperature effect (Yu et al., 2016; Teng et al., 2017).

### 3. Results

**3.1 Temporal variations in the concentrations of basic gases and their $PM_{2.5}$ counterparts in the coastal atmosphere**

Before analyzing the basic gases and their counterparts in the marine atmosphere, we first presented their continental

concentrations at the coastal site facing the Yellow Sea, as these observations provide important evidence to facilitate the analysis of the contributors to these species in the marine atmosphere. Figures 2a & b show that the $TMA_{gas}$ and $TMAH^+$ concentrations in $PM_{2.5}$ always approached the detection limit, varying at approximately $0.002\pm0.001$ µg m$^{-3}$ (average ± standard deviation), regardless of the presence of offshore or onshore winds. The $DMA_{gas}$ and $DMAH^+$ concentrations varied at $0.017\pm0.023$ and $0.017\pm0.012$ µg m$^{-3}$, respectively, which were approximately one order of magnitude larger than those of $TMA_{gas}$ and $TMAH^+$. This suggests that the $TMA_{gas}$ and $TMAH^+$ concentrations in the upwind continental and coastal atmospheres were extremely low. However, this was not the case–five to ten years ago. For example, the concentrations of the two aminium ions were comparable in atmospheric particles collected at two other coastal sites located approximately 20 km from the study area (Yu et al., 2016; Xie et al., 2018). The cause of this change is beyond the scope of this study, but may be the large decrease in manure application, based on our recent survey in the Qingdao area.

The $DMA_{gas}$ and $DMAH^+$ in $PM_{2.5}$ concentrations with offshore winds were substantially higher than those with onshore winds, suggesting that their continental emissions and related secondary sources were stronger. Moreover, the concentrations of $DMA_{gas}$ and $DMAH^+$ were moderately correlated with those of $NH_{3gas}$ and $NH_4^+$, i.e., $[DMA_{gas}] = 5.1\times10^{-3}\times[NH_{3gas}]$ ($R^2=0.69$, $P<0.01$), and $[DMAH^+]_{PM2.5}=6.1\times10^{-3}\times [NH_4^+]_{PM2.5}$ ($R^2=0.66$, $P<0.01$). Generally, the $DMA_{gas}$ and $DMAH^+$ concentrations were approximately 1/200 of those of the corresponding $NH_{3gas}$ and $NH_4^+$.

### 3.2 Spatiotemporal variations in the concentrations of basic gases over the seas

Throughout Campaign A, the $TMA_{gas}$ concentrations varied at approximately $0.031\pm0.009$ µg m$^{-3}$ (Fig 1a-c), with three peaks occurring at 4- to 5-d intervals (gray shadowing in Fig. 1c). Peaks 1 and 2 were generally associated with offshore winds, while Peak 3 was mostly associated with onshore winds (Fig. 1b). The peaks lasted from tens to dozens of hours and were not caused by onboard dew evaporation at sunrise. For example, the highest value ($0.060$ µg m$^{-3}$) occurred at 23:00 on December 16. The observed concentrations were one order of magnitude higher than those measured in the coastal atmosphere during the summer. The values were also significantly higher than those of $DMA_{gas}$ ($P<0.01$), which varied at approximately $0.006\pm0.006$ µg m$^{-3}$ (Fig 1d). The comparison results strongly indicated that the $TMA_{gas}$ observed during Campaign A was largely derived from marine sources. The same conclusion could be drawn by analyzing the three peaks of

$TMA_{gas}$ and its temporal variations during the anchoring port period. For example, during Peak 1 (Fig. 1a), the concentrations of $TMA_{gas}$ increased by approximately 100% from 20:00 on December 9 to 11:00 on December 10 with a decrease in the $SO_4^{2-}$ concentration of approximately 30% (from 23 to 17 μg m$^{-3}$; Fig 1b). Moreover, the peaks in the $TMA_{gas}$ concentrations corresponded to troughs in the $SO_4^{2-}$ concentrations during Peak 3, as shown in Figs 1c & d. The self-vessel

emissions of $SO_4^{2-}$ in $PM_{2.5}$ were negligible due to the use of low-sulfur diesel, which will be discussed later. The increased $SO_4^{2-}$ concentrations of $PM_{2.5}$ may be a good indicator of continental transport, and vice versa.

Unlike $TMA_{gas}$, continental transport likely acted as an important contributor to the $DMA_{gas}$ and $NH_{3gas}$ observed in the marine atmosphere, particularly during Peak 1, when higher $SO_4^{2-}$ concentrations were observed in $PM_{2.5}$ (Figs 1c-e). The $DMA_{gas}$ and $NH_{3gas}$ concentrations were negatively correlated with those of $TMA_{gas}$ during Peak 1, suggesting that most of

the $DMA_{gas}$ and $NH_{3gas}$ were likely derived from continental transport, rather than marine sources. During Peak 2, increased $TMA_{gas}$, $DMA_{gas}$, and $NH_{3gas}$ concentrations were observed concurrently with increasing $SO_4^{2-}$ concentrations, suggesting that both the marine emissions and continental transport may contribute to the observed $DMA_{gas}$ and $NH_{3gas}$ at the same moment. During the port-anchoring period on 20-22 December, the $DMA_{gas}$ and $NH_{3gas}$ concentrations varied slightly, and were moderate and low, respectively. However, the $TMA_{gas}$ concentrations continuously increased by over 100% as the

ambient temperature increased (Figs 1c and f). Additionally, the $SO_4^{2-}$ concentrations of $PM_{2.5}$ varied greatly and followed a bell-shaped pattern during the port-anchoring period.

Additionally, the $NH_{3gas}$ concentrations varied at approximately $0.53 \pm 0.53$ μg m$^{-3}$ from December 9 to 22. The variation narrowed to approximately $0.24 \pm 0.07$ μg m$^{-3}$ during the port-anchoring period on December 19-22. When the data during Campaign A were used for analysis, the $NH_{3gas}$ concentrations were significantly correlated with those of $DMA_{gas}$; i.e.,

$[DMA_{gas}] = 9.3 \times 10^{-3} \times [NH_{3gas}]$ ($R^2$=0.35, P<0.01). However, there was no correlation between the $NH_{3gas}$ and $TMA_{gas}$ concentrations.

### 3.3 Spatiotemporal variations in the aminium and $NH_4^+$ ion concentrations of $PM_{2.5}$ over the seas

Figures 3a-f show the spatiotemporal variations in the $TMAH^+$, $DMAH^+$, and $NH_4^+$ concentrations of $PM_{2.5}$ throughout Campaign A from December 9 to 22, during which the $TMAH^+$ concentrations varied greatly at approximately $0.28\pm0.18$ μg



m$^{-3}$. However, they narrowed at approximately 0.21±0.04 μg m$^{-3}$ during the port-anchoring period. The TMAH$^+$ concentrations generally increased from 0.13±0.05 μg m$^{-3}$ on December 9 to 0.46±0.05 μg m$^{-3}$ on December 16 (Fig. 3a), and then decreased to approximately 0.2 μg m$^{-3}$ afterward, excluding some strong peaks of 0.62–1.24 μg m$^{-3}$ at 03:00–05:59 and 1.02–1.81 μg m$^{-3}$ at 14:00–16:59 on 18 December (grey shadowing as Peak 4 in Figs 3a-d). The peaks reproduced the episodes observed in the marine atmosphere over the Yellow Sea during May 2012 (Hu et al., 2015) and were repeatedly

observed during Campaign B (Gao et al., submitted to ACP), but were not observed in the several other marine cruise campaigns conducted across the marginal seas of China and northwest Pacific Ocean (Hu et al., 2018; Xie et al., 2018).

As the TMAH$^+$ concentrations were approximately two orders of magnitude higher than the observations at the coastal site during summer 2019, the observed TMAH$^+$ were likely largely derived from marine sources. The TMAH$^+$ concentrations followed a spatiotemporal pattern that was clearly different from that of DMAH$^+$ and NH$_4^+$, while the latter two ions

exhibited a similar spatiotemporal pattern during most of the periods throughout Campaign A (Figs 3a-c). A significant negative correlation (P<0.01) was obtained between the concentrations of TMAH$^+$ and NH$_4^+$ in PM$_{2.5}$ (not shown). The spatiotemporal pattern of the TMAH$^+$ concentration was also greatly different to those of SO$_4^{2-}$ (Fig. 1d) and SO$_2$ (Fig. 3b). For example, the extremely strong TMAH$^+$ peaks occurred concurrently with low SO$_4^{2-}$, NH$_4^+$, and SO$_2$ concentrations, while accompanying with high concentrations of Na$^+$ under high wind speeds as indicators of sea spray aerosols (Feng et al., 2017).

Moreover, the TMAH$^+$ concentrations were approximately one order of magnitude larger than those of TMA$_{gas}$, and no significant correlation was observed between them (P>0.05). This suggests that the observed TMAH$^+$ may not be derived from the neutralization reactions of TMA$_{gas}$ with acids in the marine atmosphere, and may have been derived from primary sea-spray organic aerosols (Ault et al., 2013; Prather et al., 2013; Quinn et al., 2015; Hu et al., 2018; Dall'Osto et al., 2019).

The DMAH$^+$ concentrations varied at approximately 0.065±0.068 μg m$^{-3}$ on December 9-22; however, they varied at

approximately 0.10 ±0.04 μg m$^{-3}$ during the port-anchoring period. The 25th percentile value of DMAH$^+$ during Campaign A was 0.021 μg m$^{-3}$, suggesting a low background concentration in the marine area. The DMAH$^+$ concentrations were significantly correlated with those of NH$_4^+$ (R$^2$=0.71, P<0.01; data not shown). When the data obtained at 03:00–05:59 and 14:00–16:59 on December 18 (strong peaks of TMAH$^+$ with a simultaneous increase in DMAH$^+$) were removed for correlation, the R$^2$ value improved to 0.78. Unlike the TMAH$^+$, the observed DMAH$^+$ may have been partially derived from

acid-basic neutralization reactions in ambient air, in addition to the primary sea-spray organic aerosols. For example, largely

increased DMAH$^+$ concentrations occurred concurrently with strong peaks in the TMAH$^+$ concentrations (gray shadowed

peak 4 in Figs 3a & b).

The NH$_4^+$ concentrations of PM$_{2.5}$ varied greatly at approximately 4.7±7.2 μg m$^{-3}$ during Campaign A (Fig. 3c). However,

the 25$^{th}$ percentile values were as low as 0.21 μg m$^{-3}$, suggesting low marine background values. The 50$^{th}$ percentile value

was also only 1.2 μg m$^{-3}$, which was much smaller than the average value due to the presence of strong peaks in the NH$_4^+$

concentrations. The increased NH$_4^+$ concentrations associated with NO$_3^-$ and SO$_4^{2-}$ during Campaign A were likely due to

long-range transport from the upwind continents.

## 4. Discussion

### 4.1 Effects of temperature on the observed basic gases in the marine atmosphere

As mentioned above, the observed TMA$_{gas}$ likely originated from marine sources. We plotted the concentrations of TMA$_{gas}$

against the ambient air temperature (T) in Fig. 4a, which generally increased with increasing T. We further separated the

average hourly wind speeds (WS) into three categories, i.e., WS ≤ 5.0, 5.0< WS ≤9.0, and WS>9.0 m s$^{-1}$. At WS>9.0 m s$^{-1}$,

the data obtained from 15:00 on December 16 to 01:00 on December 19 including Peaks 3 and 4, were separately considered

as half-full symbols in Fig. 4a. The TMA$_{gas}$ concentrations (half-full symbols) generally exceeded the concentrations of the

other gases at the same T, with which they exhibited a moderately good exponent correlation, ([TMA$_{gas}$] = 0.03 × e$^{0.04T}$ with

R$^2$=0.72). From 15:00 on December 16 to 01:00 on December 19 stronger emission potentials of TMA$_{gas}$ to the marine

atmosphere were expected in the corresponding marine zone. Although the TMAH$^+$ in the surface seawater was not directly

measured, higher TMAH$^+$ concentrations were expected.

Following the same approach, the DMA$_{gas}$ and NH$_{3gas}$ concentrations were plotted against T, as shown in Figs 4b & c,

respectively. They generally increased with increasing T. The NH$_{3gas}$ concentrations (half-full symbols) were extremely well

correlated with T ([NH$_{3gas}$] = 0.05 × e$^{0.3T}$ with R$^2$=0.96). As lower concentrations of SO$_4^{2-}$, NH$_4^+$, and SO$_2$ were generally

observed at the same time, the continental transport of NH$_{3gas}$ should be greatly reduced; therefore, the observed NH$_{3gas}$ was

likely mainly derived from the seas. Therefore, the seas were the net source of NH$_{3gas}$ at the measurement time. However, at



the same T, the $NH_{3gas}$ concentrations (half-full symbols) were generally lower than those during the other periods in this

study. The concentrations of $NH_4^+$ in the surface seawater may have been lower at the measurement time. However, this may

not be the case as higher concentrations of $TMAH^+$ were expected. Alternatively, the continental transport of $NH_{3gas}$ may

have made an important contribution to the observed $NH_{3gas}$ during most of the other periods when the seas were the net

$NH_{3gas}$ sink.

$DMA_{gas}$ exhibited an extremely good exponent correlation with T (half-full symbols) at the measurement time ($[NH_{3gas}] =$

$0.002 \times e^{0.3T}$ with $R^2=0.93$). At the same T, the $DMA_{gas}$ concentrations (half-full symbols) were not always higher or lower

than the others. Two scenarios were considered. Under Scenario 1, the observed $DMA_{gas}$ concentrations exceeded the values

predicted by the regression equation using the ambient T as the input. The seas were the likely net sinks of the $DMA_{gas}$.

Under Scenario 2, including all others, measurements of the $DMAH^+$ in the surface seawater were required to confirm

whether the seas were the net sources or sinks of $DMA_{gas}$.

**4.2 Estimating sea-derived $DMA_{gas}$ and $NH_{3gas}$ in the marine atmosphere**

To estimate the sea-derived $DMA_{gas}$ and $NH_{3gas}$ concentrations in the marine atmosphere, we plotted the $DMA_{gas}$ and $NH_{3gas}$

concentrations against $TMA_{gas}$, as shown in Figs 5a & b. The purple-red and dark-green markers represent the data obtained

during the increasing and decreasing periods of Peak 3, respectively, which were analyzed separately. A good correlation can

be obtained between $DMA_{gas}$ and $TMA_{gas}$ during the increasing period ($[DMA_{gas}] = 0.63\times[TMA_{gas}] - 0.01$, $R^2=0.89$ and

$P<0.01$). The equation for the decreasing period was as follows: $[DMA_{gas}] = 1.3\times[TMA_{gas}] - 0.05$, $R^2=0.79$ and $P<0.01$.

Therefore, the $TMAH^+$ in the surface seawater may decompose into $DMAH^+$ to different extents (Lidbury et al., 2015a;

Lidbury et al., 2015b; Xie et al., 2018). The two regression curves (purple-red and dark-green dashed lines in Figs 5a & b)

created a large triangular zone that likely reflected the different ratios of $DMA_{gas}/TMA_{gas}$ in primary marine emissions. We

assumed that any data beyond the purple-red dashed line reflected the contribution of non-sea-derived $DMA_{gas}$, which should

be attributed to continental transport. Therefore, we assumed that the non-sea-derived $DMA_{gas}$ ($DMA_{gas}^{\#}$) concentrations

were equal to the observed values of $DMA_{gas}$ minus the predicted values obtained using $[DMA_{gas}] = 0.63\times[TMA_{gas}] - 0.01$,

and the calculated $DMA_{gas}^{\#}$ values are shown in Fig. 5c. Based on the triangular zone in Fig. 5a, the calculated values should





be considered as the lower limit of $DMA_{gas}^{\#}$. During Peak 1, the calculated $DMA_{gas}^{\#}$ contributed to over 40% of the observed $DMA_{gas}$ for 12 h. Similar calculated results for $DMA_{gas}^{\#}$ were obtained during Peak 2.

The same approach was employed to analyze the $NH_{3gas}$ results, as shown in Figs 5b and d. During Peak 1, the calculated non-sea-derived $NH_{3gas}$ ($NH_{3gas}^{\#}$) contributed to over 40% of the observed $NH_{3gas}$ for 17 h. During Peak 2, the calculated $NH_{3gas}^{\#}$ contributed to over 40% of the observed $NH_{3gas}$ for 24 h.

Overall, the $DMA_{gas}^{\#}$ and $NH_{3gas}^{\#}$ concentrations varied at approximately $0.001\pm0.003$ and $0.18\pm0.39$ µg m$^{-3}$, respectively. The calculated average $DMA_{gas}^{\#}$ and $NH_{3gas}^{\#}$ values accounted for 16% and 34% of the observed averages of each species,

respectively.

### 4.3 Estimation of non-sea-spray particulate DMAH$^{+}$ in the marine atmosphere

We plotted the concentrations of DMAH$^{+}$ against those of TMAH$^{+}$ in PM$_{2.5}$ (Fig. 6a) using the data obtained from 15:00 on December 16 to 01:00 on December 19 ( $[DMAH^{+}]_{PM2.5} =0.13 \times [TMAH^{+}]_{PM2.5}$, $R^{2}=0.91$, $P<0.01$). We assumed that the non-sea-primarily derived DMAH$^{+}$ concentrations in PM$_{2.5}$, marked as DMAH$^{+\#}$, were equal to the observed DMAH$^{+}$ values

minus the predicted values using the regression equation. The calculated DMAH$^{+\#}$ values are shown in Fig. 6b. The DMAH$^{+\#}$ concentrations varied at approximately $0.042\pm0.070$ µg m$^{-3}$ throughout Campaign A, during which the calculated average DMAH$^{+\#}$ accounted for 65% of the observed average. Additionally, the calculated DMAH$^{+\#}$ values accounted for over 80% of the observed values in 26% of the Campaign-A period. Again, the decomposition of TMAH$^{+}$ to DMAH$^{+}$ may have occurred in surface seawater and/or the marine atmosphere, to an extent, and the estimated DMAH$^{+\#}$ should be

considered as the upper limit. Note that the NH$_{4}^{+}$ and TMAH$^{+}$ concentrations were negatively correlated during Campaign A, and no primary particulate NH$_{4}^{+}$ from sea-spray aerosols could be identified.

When the concentrations of DMAH$^{+\#}$ were plotted against those of NH$_{4}^{+}$ (Fig 6c), we obtained the following equation: $[DMAH^{+\#}]_{PM2.5} = 0.0089\times [NH_{4}^{+}]_{PM2.5}$, $R^{2}=0.82$, $P<0.01$. The slope was larger than that obtained in the coastal atmosphere during the summer (0.0061). This difference may be partially explained by the gas-aerosol equilibria among them (Pankow,

2015; Xie et al., 2018), considering the two regression equations, i.e., $[DMA_{gas}] = 9.3 \times 10^{-3} \times [NH_{3gas}]$ in the marine atmosphere, and $[DMA_{gas}] = 5.1 \times 10^{-3} \times [NH_{3gas}]$ in the summer coastal atmosphere.



**4.4 Formation and chemical conversion of aminium ions in the transported and self-vessel SO₂ plumes**

When the sea-spray particulate DMAH$^+$ was deducted, the increased concentrations of DMAH$^{+\#}$ were generally associated with increased SO$_4^{2-}$ and SO$_2$ concentrations. Combining this with the moderate correlation between DMAH$^{+\#}$ and NH$_4^+$, it

can be inferred that the DMAH$^{+\#}$ likely originated from concurrent secondary formation with NH$_4^+$. However, we separated the air pollutant plumes into two groups. Group 1 represented an increase in SO$_4^{2-}$ and NH$_4^+$ together with SO$_2$, while Group 2 represented an increase in SO$_2$ without increases in SO$_4^{2-}$ and NH$_4^+$. Group 1 likely reflected the transport of aged air pollutant plumes from the continents, while Group 2 may reflect self-vessel SO$_2$ plumes. As shown in Figs 6b and 3b-c, the concentrations of DMAH$^{+\#}$ and NH$_4^+$ in the self-vessel SO$_2$ plumes did not increase in the intervals between Peaks 1 and 2,

and between Peaks 2 and 3. Therefore, no fresh formation of DMAH$^{+\#}$ and NH$_4^+$ in the self-vessel emissions was detected. However, the concentrations of TMAH$^+$ decreased in some self-vessel SO$_2$ plumes. The TMAH$^+$ concentrations were approximately one order of magnitude higher than those of TMA$_{gas}$ in the marine atmosphere. Assuming that the decreased TMAH$^+$ was released from PM$_{2.5}$ to the gas phase, a simultaneous large spike in TMA$_{gas}$ should be observed. However, this was not the case, as shown in Fig 1c. The decreased TMAH$^+$ may persist in the PM$_{2.5}$, but could not be detected by AIM-IC.

**4. Conclusion and Implication**

In continental China upwind of the Yellow Sea, the TMA$_{gas}$ and TMAH$^+$ concentrations in PM$_{2.5}$ were extremely low (0.002±0.001 µg m$^{-3}$), close to the detection limit of the AIM-IC. Taking the observations as a reference, the largely increased TMA$_{gas}$ (0.031±0.009 µg m$^{-3}$) and particulate TMAH$^+$ (0.28±0.18 µg m$^{-3}$) concentrations in the marine atmosphere were attributed to marine emissions. Therefore, TMA$_{gas}$ and particulate TMAH$^+$ can be used as unique tracers to quantify the

marine emissions of DMA$_{gas}$, NH$_{3gas}$, and particulate DMAH$^+$, as well as the long-range transport from upwind continental China.

Through comprehensive comparison and correlation analyses, the high concentrations of TMAH$^+$ in PM$_{2.5}$ observed over the Yellow and Bohai Seas, with episodic average hourly exceeding over 1 µg m$^{-3}$, were inferred to originate from strong primary sea-spray aerosol emissions. Moreover, the TMA$_{gas}$ concentrations generally increased with increasing ambient

temperature and sea surface wind speeds, suggesting that the observed TMA$_{gas}$ was likely released from the surface seawater.

However, the $TMA_{gas}$ concentrations were substantially lower than those of particulate $TMAH^+$, and were not significantly correlated. Although different mechanisms of the release of $TMA_{gas}$ and particulate $TMAH^+$ from the seas have been reported in the literature, the lack of a significant correlation between them was surprising and is explored in the companion study.

The $DMA_{gas}$ and $NH_{3gas}$ concentrations varied at approximately $0.006\pm0.006$ and $0.53\pm0.53$ µg m$^{-3}$ during Campaign A, in which at least 16% and 34 % of the observational values were derived from continental transport, respectively. The sea-derived $DMA_{gas}$ and $NH_{3gas}$ were likely released with $TMA_{gas}$ as they peaked simultaneously. The $DMAH^+$ concentrations of $PM_{2.5}$ varied at approximately $0.065\pm0.068$ µg m$^{-3}$ during Campaign A, 65% of which was derived from continental transport.

Our analysis results did not support the occurrence of the photolysis of marine organic nitrogen to generate $NH_{3gas}$ in the marine atmosphere during winter as there was no correlation between the sea-derived $NH_{3gas}$ and particulate $TMAH^+$ concentrations. Additionally, Peaks 2 and 3 of $NH_{3gas}$ persisted for dozens of hours under strong winds and were unlikely to be derived from seabird emissions. Alternatively, a good exponent correlation was observed between the observed $NH_{3gas}$ concentrations and T during the period lacking continental air pollutant transport, suggesting that the observed $NH_{3gas}$ was 305   released from seawater. $NH_3$ emissions via seabirds were unlikely to be an important contributor to the observed $NH_{3gas}$ in the marine atmosphere during winter, although this may not have been the case during other seasons.

Additionally, no formation of particulate $NH_4^+$ and $DMAH^+$ in the self-vessel $SO_2$ plume was observed in the marine atmosphere. However, the particulate $TMAH^+$ concentration clearly decreased in the self-vessel $SO_2$ plume without a simultaneous increase in the $TMA_{gas}$ concentrations. Undetectable chemical conversion of particulate $TMAH^+$ by AIM-IC 310   likely occurred and requires further investigation.

*Data availability*. The data of this paper are available upon request (contact: Xiaohong Yao, xhyao@ouc.edu.cn).

**Acknowledgment**

This research is supported by the National Key Research and Development Program in China (grant no. 2016YFC0200504), the Natural Science Foundation of China (grant no. 41776086)



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





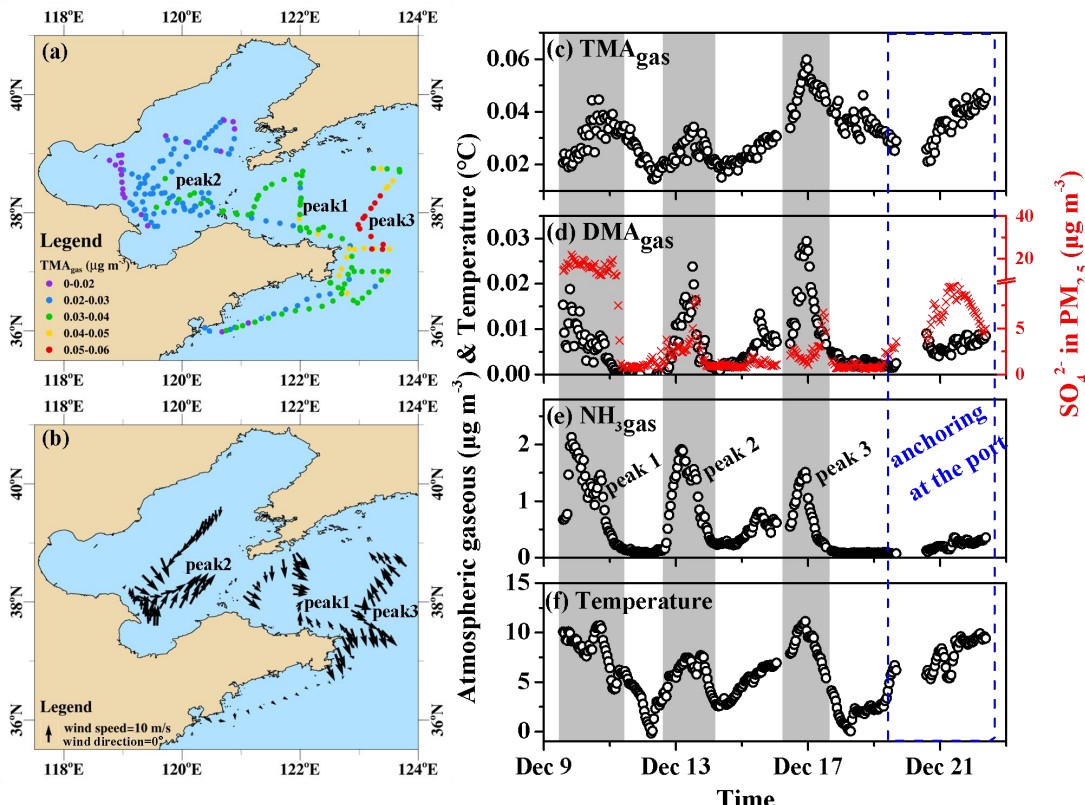


**Figure 1: Spatiotemporal variations in the concentrations of basic gases and other parameters during the Yellow and Bohai Sea cruise campaigns on December 9-22, 2019 (mapping TMA$_{gas}$ by concentration (a); mapping onboard recorded wind speeds and directions (b); time-series of TMA$_{gas}$ (c), DMA$_{gas}$, (d), NH$_{3gas}$ (e), and onboard recorded ambient air temperature (f); the time-series of SO$_4^{2-}$ in PM$_{2.5}$ were shown as indicators of anthropogenic air pollutants in (d); not all data were shown in (b) to avoid**

**clustering).**

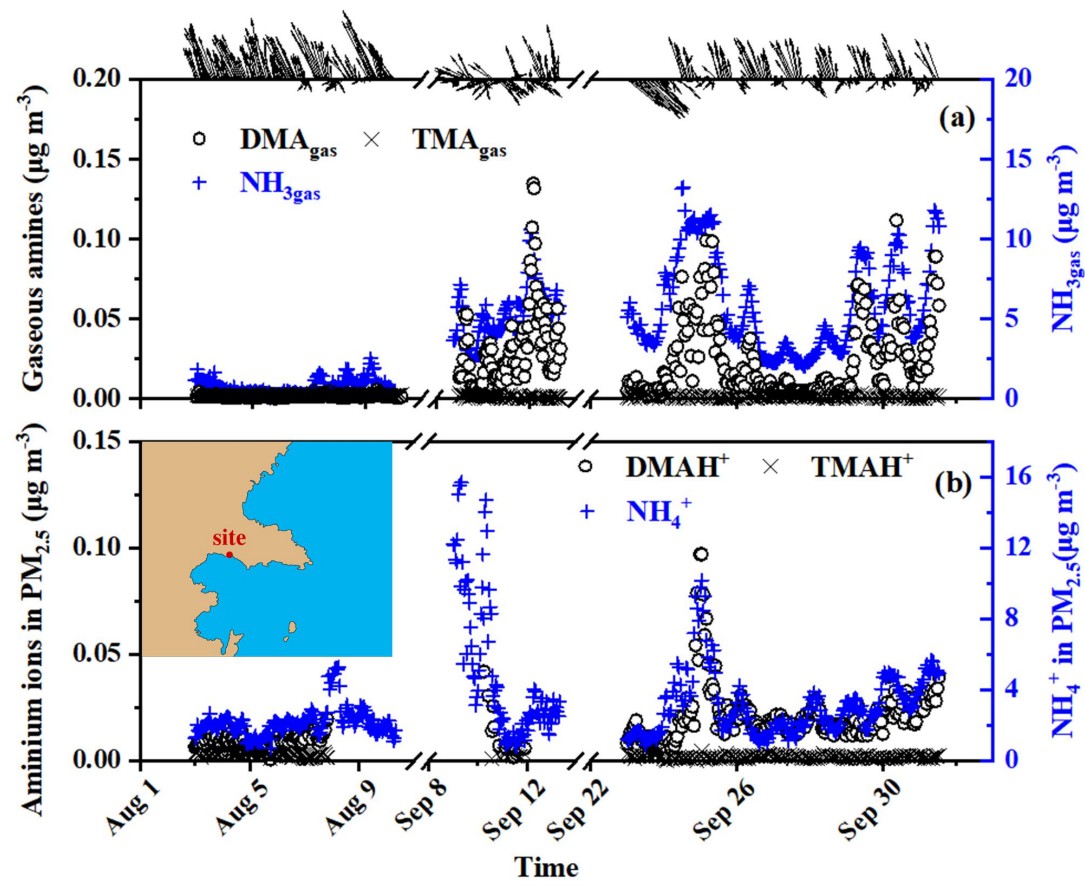

**Figure 2: Temporal variations in the concentrations of NH$_{3gas}$ and gaseous amines and their counterparts in PM$_{2.5}$ at a coastal site during August and September 2019 (NH$_{3gas}$ and gaseous amines (a); counterparts in PM$_{2.5}$ (b); wind speed and direction superimposed on the top of (a); a map of the sampling site superimposed in (b); the missing data regarding aminium ions in PM$_{2.5}$ were due to occasional K$^+$ contamination (b)).**






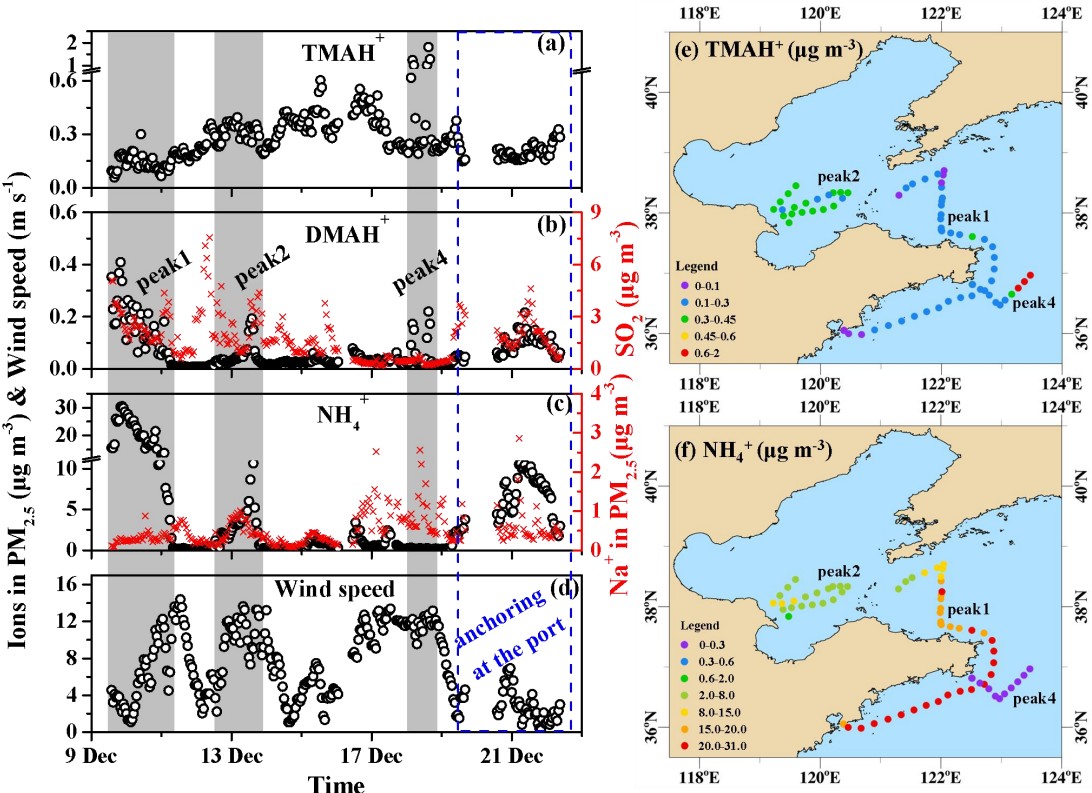

**Figure 3: Spatiotemporal variations in the aminium ions and NH$_4^+$ concentrations of PM$_{2.5}$ and other parameters during the cruise campaign over the Yellow and Bohai Seas on 9-22 December 2019 (time-series of TMAH$^+$ (a), DMAH$^+$ (b), and NH$_4^+$ in PM$_{2.5}$ (c), wind speeds (WS) (d); mapping of the TMAH$^+$ in concentration (e); mapping of the NH$_4^+$ concentration (f); the time-series of SO$_2$ are shown as an indicator in (b); the time-series of Na$^+$ in PM$_{2.5}$ were shown as an indicator of sea spray aerosols in (c); only some data were used in (e) and (f) to avoid clustering)**


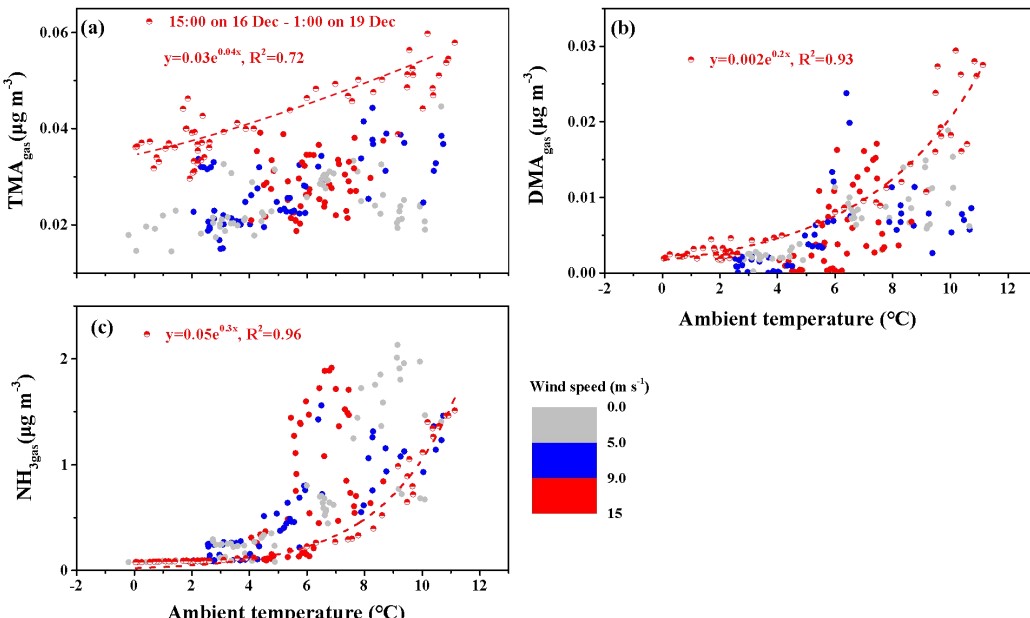

**Figure 4: Correlations between the concentrations of basic gases and ambient temperature (TMA$_{gas}$ (a); DMA$_{gas}$ (b); and NH$_3$ (c); the colored bar represents different wind speeds; full symbols represent the data observed throughout the campaign excluding the period from 15:00 on December 16 to 01:00 on December 19).**





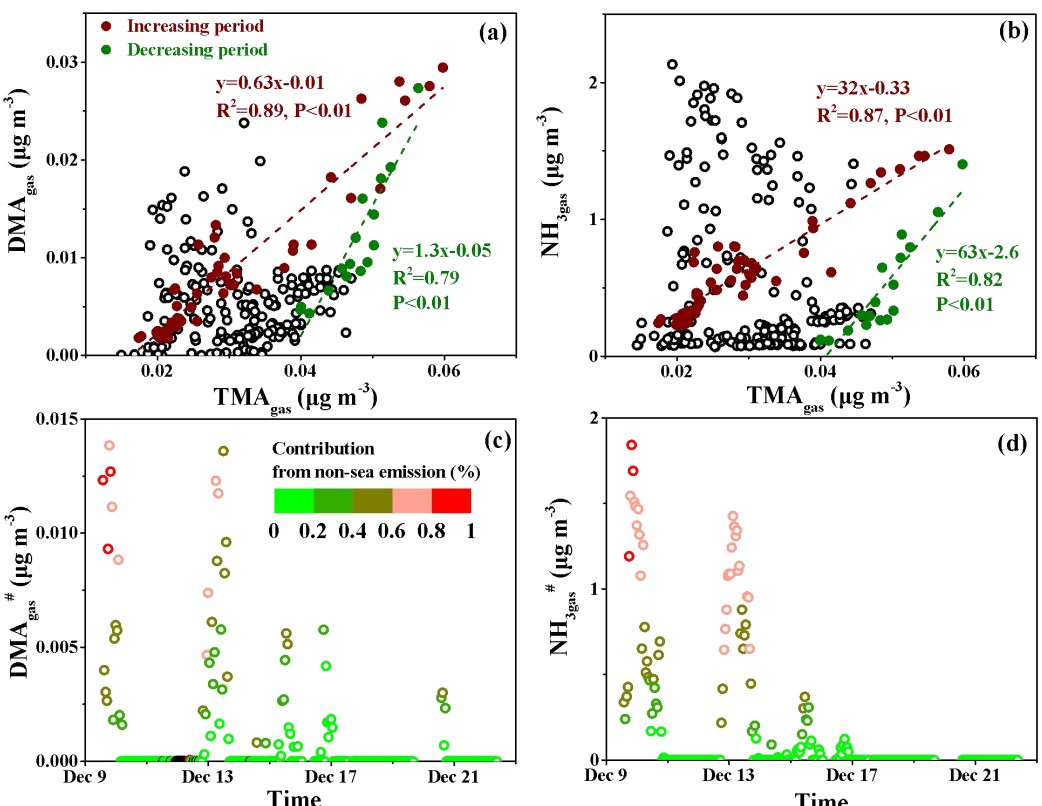

**Figure 5: Correlations of DMA$_{gas}$ and NH$_{3gas}$ with TMA$_{gas}$ and time-series of the calculated DMA$_{gas}$# and NH$_{3gas}$# (DMA$_{gas}$ vs TMA$_{gas}$ (a); NH$_{3gas}$ vs TMA$_{gas}$ (b); DMA$_{gas}$# (c); and NH$_{3gas}$# (d); the colored bars in (c) and (d) represent the percentages of transported DMA$_{gas}$# and NH$_{3gas}$# in each corresponding observed value).**



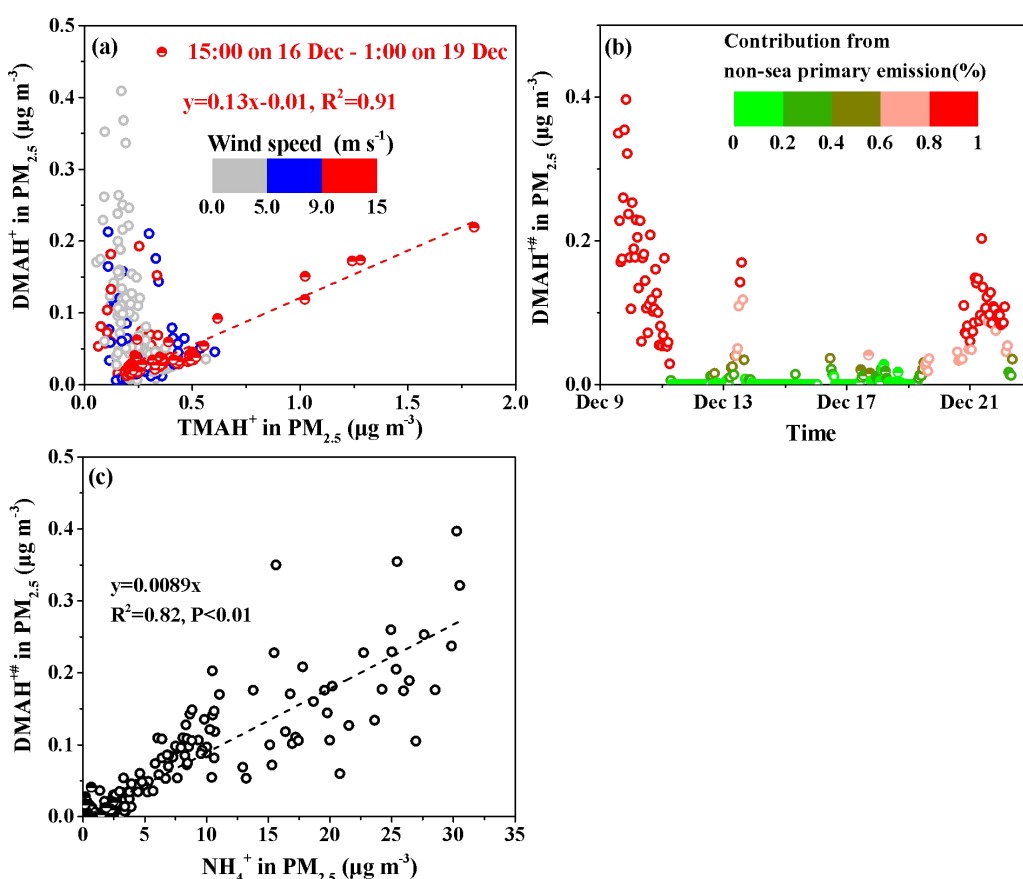


Figure 6: Correlations analyses of different variables in PM$_{2.5}$ and time-series of the calculated DMAH$^{+\#}$ in PM$_{2.5}$ (DMAH$^+$ vs TMAH$^+$ (a); time-series of DMAH$^{+\#}$ (b); DMAH$^{+\#}$ vs NH$_4^+$; DMAH$^+$ vs NH$_4^+$).