# Peer review of "Mapping gaseous dimethylamine, trimethylamine, ammonia, and their particulate counterparts in marine atmospheres of China's marginal seas: Part 1 - Differentiating marine emission from continental transport"

_Atmospheric Chemistry and Physics, 2021_

## Author Response (AR1)

August 27, 2021

Associate Editor, Dr. Maria Kanakidou

**Response Letter: Revision of Manuscript # acp-2021-258**

Dear Dr. Maria Kanakidou:

We have made revisions according to the comments. Here is a point-by-point summary of our response to comments and suggestions. The responses have been revised according to the final revision of the manuscript. The comments are listed first, and our responses follow each comment. We also checked and revised the whole manuscript and figures.

Best regards!

Sincerely,
Xiaohong Yao, Ph.D.
Ocean University of China

The responses have been revised according to the final revision of the manuscript.

**Response to comments by Anonymous Referee #1**

*(1) The data presented in this paper could add to our understanding of TMA and DMA in gas and in particulate phase, in China's marginal seas. This should be also reflected in the title since the word "amines" is misleading because it is discussed only two compounds. Furthermore, some technical details, especially in the analytical protocol, are not presented adequately.*

**Response:** Agree. The title has been changed to "Mapping gaseous dimethylamine, trimethylamine, ammonia, and their particulate counterparts in marine atmospheres of China's marginal seas: Part 1 - Differentiating marine emission from continental transport".

The technical details of our analytical protocol have been added in the revised experimental section.

*(2) The different sections have not the correct numbers e.g. the section "4. Conclusions" should be 5, while the sections "Introduction" and "Experimental" do not have numbers.*

**Response:** Sorry for this. It has been corrected in the revision.

*(3) In the Experimental section: I propose to split it into the sampling and analysis part.*

**Response:** Agree. Revised.

*(4) The introduction is focused on the biogenic sources of amines. What about anthropogenic emissions? Is there any information from the literature, especially for the area?*

**Response:** In addition to biogenic emissions of amines, anthropogenic emissions have been reviewed as important sources of amines in the continental atmosphere, but not in the marine atmosphere (Ge et al., 2011). Modeling studies showed that the continental amine species in gas and/or particle phases can be transported regionally, including downwind marine atmospheres (Yu and Luo, 2014; Mao et al., 2018). This has been also added in the revision.

*(5) Lines 114-118. More details regarding the analysis should be added here, such as the eluent, the type and time of elution, injection volume, the detection limit for anions.*

**Response:** Agree. Added.

*(6) Line 123. What exactly the author means by the term contamination of $K^+$? Is it related to the overlapping of the peaks? If yes, the term "interference" is more appropriate. I propose to present a chromatograph in the supplement material with and without $K^+$ interference.*

**Response:** Thanks. The word "interference" should be better and changed accordingly. Figures with and without $K^+$ interference have been added in the supplement material.

*(7) Line 174. What are those values?*

**Response:** Those values are referred to "The observed concentrations of $TMA_{gas}$", which have been clarified in the revision.

*(8) Line 187. "Negatively correlated" please refer $r^2$ and p.*

**Response:** $R^2$ and p value have been given in the revision.

(9) Line 221. $SO_4^{2-}$ anions are non-sea salt? If not, it should be clarified for continental sources.

**Response:** $Nss\text{-}SO_4^{2-}$ instead of $SO_4^{2-}$ has been used for the tracer of continental sources in the revised Figure 2(d) and through the revised manuscript.

*(10) Line 223. Na is sea salt Na? If not it should be for marine sources.*

**Response:** We had no additional data to confirm sea-salt Na. However, $Na^+$ in $PM_{2.5}$ over the marine atmosphere far away from the continents should be mainly derived from marine sources, especially the increased concentrations of $Na^+$ in $PM_{2.5}$ with increasing wind speeds at 10 m s$^{-1}$. In the revision, the sentence has changed to "while accompanying with high concentrations of $Na^+$ under high wind speeds as commonly assumed to be indicators of sea spray aerosols (Feng et al., 2017)."

*(11) It would be nice to change the order of Figures 1 and 2, in order to help the reader to follow the discussion.*

**Response:** Agree. Done.

*(12) Fig. 2 (b) where is wind speed? Please explain how the direction of arrows is connected with the wind direction.*

**Response:** Agree. The length of arrows was superimposed on the top of (a) represents wind speed. We have added a legend for reference in Fig. 2(a) to shown the speed and direction of wind.

*(13) Fig.3. The legends are missing from the graph.*

**Response:** Added in the revision.

*(14) Fig.3. "only some data were used in (e) and (f) to avoid clustering". Which was the criterion of "some data"?*

**Response:** In the revision, we added "to better show spatiotemporal distributions of TMAH$^+$ and NH$_4^+$ during Peak 1, Peak 2 and Peak 4, only data during periods shaded in (a-d) were used in (e) and (f) to avoid clustering."

*(15) Fig.4. Remove the single bracket.*

**Response:** Sorry. We can't get the point.

The responses have been revised according to the final revision of the manuscript.

**Response to comments by Anonymous Referee #2**

*(1) Between lines 140-145, the comparison was made between two different time periods and this might be meaningless unless the authors can provide strong rationale to do so. Also, conclusions made on the origin of the particulate $TMAH^+$ between lines 177-178 were based on similar comparison. Can the authors be sure that the $TMAH^+$ concentration was highest in summer than in any other seasons?*

**Response:** The measurement data at the same coastal site during the period on November 15 to December 1 in 2019, two weeks before the campaign, were added in Fig 1 as additional measurement collected at the coastal site near the Yellow Sea together with data collected during summer and fall 2019. The data also showed that the concentrations of $TMA_{gas}$ were extremely low and mostly below the limit of detection. The corresponding concentrations of $TMAH^+$ observed were also extremely low and mostly undetectable during the cold period. Both the summer and the cold season data allow to draw the conclusion, i.e., the continental transport of $TMA_{gas}$ and particulate $TMAH^+$ to downwind marine atmosphere was likely negligible.

*(2) What are the backgrounds or interferences for both $NH_3$ and $NH_4^+$, the gaseous amines, aminum from the URG and potentially affect the measurement results?*

**Response:** We speculate that the comments related to the deputy loop volume installed on the low-pressure injection valve. The limits of detection (LOD) for ions in AIM-IC are largely adjustable and almost completely determined by the user-chosen volume of the loop. We adopted the loop volume at 250 -2000 μL for measurements in different atmospheres to lower the LOD and gain reasonably accurate concentrations of these species. In this study, 250 μL loop was used by considering that high $NH_{3gas}$ concentrations may occur in the marine atmosphere. The information has been added in the revision.

In our measurements, the major interference to aminium measurements was derived from the unexpected occurrence of $K^+$ interference. This had been mentioned in the method section and more data has been added in the revision.

Since 2014, we tried to measure $NH_{3gas}$ and amine gases in various marine atmospheres. However, no reasonable data can be gained until 2018. We combined multiple $NH_3$ gas analyzers and off-line denuder samplings during two cruise campaigns in 2016-2017, and found dew evaporation on the vessel surface with sunrise in the morning to be the major positive sampling artifacts of these basic gases. To solve the problem, we designed a specific container to house the AIM-IC and set the container on the front deck, where human activities were rare, to measure $NH_3$, $DMA_{gas}$, $TMA_{gas}$ and their

particulate counterparts in PM$_{2.5}$. The approach works well based on simultaneous measurements made by the AIM-IC and denuder samplers in other cruise campaigns. In addition, we made intercomparison between the AIM-IC and denuder samplers in various coastal atmospheres since 2012. We didn't find high background and other interferences, except K$^+$ interference (Teng et al., 2017).

*(3) The authors claim that the TMA and TMAH$^+$ are mainly from sea spray aerosols and hence they can be used as tracers for other basic compounds. Do the authors have any evidences between concentrations of the above two species and concentrations of sea spray aerosols or their makers? In addition, how confidence is for the measurements of the above two species?*

**Response:** Na$^+$ in PM$_{2.5}$ over the marine atmosphere far away from the continents should be mainly derived from marine sources, especially the increased concentrations of Na$^+$ in PM$_{2.5}$ with increasing wind speeds at ~10 m s$^{-1}$. In the revision, the sentence has changed to "while accompanying with high concentrations of Na$^+$ under high wind speeds as commonly assumed to be indicators of sea spray aerosols (Feng et al., 2017)."

The increased TMA$_{gas}$ with increasing wind speeds in the remote marine atmosphere is also an important evidence to release TMA$_{gas}$ through the air-sea exchange. Theoretically, increasing wind speeds in the remote marine atmosphere should decrease concentrations of TMA$_{gas}$ derived from those emissions other than the air-sea exchange.

Please see our response to Comment 2.

*(4) Rather minor: there are no section numbers for introduction and the second section.*

**Response:** Sorry for this and corrected.

The responses have been revised according to the final revision of the manuscript.

**Response to comments by Anonymous Referee #3**

*(1) My major concern lies within the question of the sensitivity of the instrument. Regarding the detection limits of the instruments and partly some time ranges where concentrations were below the LOD, I am wondering how applicable is the method for amine measurements in low concentrated (e.g. remote marine) areas. The concentrations reported here are in part significantly higher that reported in other marine regions and I think with the LODs of the here applied method, the amines would not be detectable. Please comment on the limits of the here presented technique! In this context, please compare the concentration values achieved here with literature data (especially of marine locations).*

**Response:** Thank the comments. The limits of detection (LOD) for ions in AIM-IC are largely adjustable and almost completely determined by the user-chosen volume of the loop installed on the low-pressure injection valve. We adopted the loop volume at 250 -2000 μL for measurements in different atmospheres to lower the LOD and gain reasonably accurate concentrations of these species. In this study, 250 μL loop was used by considering that high $NH_{3gas}$ concentrations may occur in the marine atmosphere. The information has been added in the revision. The reviewer's comments are valid by considering measurements made by the use of the deputy loop volume.

In the revision, more comparisons with those values reported in the literature had been added. For example, Gibb et al.(1999) measured $DMA_{gas}$ and $TMA_{gas}$ on November 16 to December 19 in 1994 over Arabian Sea, which were 8.8 ng m$^{-3}$ and 0.5 ng m$^{-3}$ respectively. The values were approximately one-two orders smaller than the concentrations of $DMA_{gas}$ and $TMA_{gas}$ measured in this work. Including the concentrations of $DMAH^+$ and $TMAH^+$ in atmospheric particles reported by Gibb et al. (1999), we also made tens of cruise off-line measurements of $DMAH^+$ and $TMAH^+$ in size-segregated atmospheric particles. Their concentrations varied largely from a few ng m$^{-3}$ to ~1 μg m$^{-3}$ in $PM_{10}$ (listed in the Table S1). The comparison has also been added in the revision.

*(2) Line 20 "we identified marine emissions of the gas species originating from continental transport…" sounds confusing. Is it marine or continental? Please clarify (and shorten the sentence)*

**Response:** We have changed the word "identified" into "differentiated".

*(3) Line 35; please add which numbers (16%, 34%) belong to which species.*

**Response:** We have added the corresponding species in the revision.

*(4) The introduction mixed amine sources in seawater and in the atmosphere. Please be more correct. For example, the first sentence of the intro states that the atmospheric amines are derived from the degradation of glycine betanine…. But the latter processes happen in the seawater (not in the atmosphere as appears from the sentence).*

**Response:** Thank you. The sentence is indeed confusing and corrected in the revision.

*(5) Line 57: Please explain, why this is not the case for the continental atmosphere.*

**Response:** The sentence has been revised as "Measuring gaseous amines in real-time simultaneously to their particulate counterparts in the marine atmosphere over the ocean remains challenging because of artifact signals related to self-vessel emissions and amine-contained dew evaporation, although this is not the case in the continental atmosphere (VandenBoer et al., 2011)."

*(6) Line 82: Do you mean, that 1) higher concentration levels of nutrients result in higher concentrations of amines?*

**Response:** This is what we expected prior to the study, yes. We have added the description for explanation. In the revision, it reads as "Winter cruise campaigns provide great opportunities for observational studies due to the 1) higher concentration levels of nutrients in the seas at a lower sea surface water temperature which may favor higher primary production (Guo et al., 2020) and subsequently increase marine emissions of gaseous amines and/or aminum-contained sea spray aerosols;"

*(7) Line 83: why is 3) "… periodically enhance long-range transport…" an advantage here?*

**Response:** In the revision, it has been revised as "periodically enhanced long-range transport of anthropogenic pollutants from continents to the seas which may enhance formation of secondary ammonium and aminium aerosols (Guo et al., 2016; Yu et al., 2016; Xie et al., 2018; Wang et al., 2019)."

*(8) Line 89: unclear expression: "identifying marine sources from continental transport…" do you mean distinguish the sources?*

**Response:** Yes and corrected to "distinguishing" in the revision.

*(9) Line 118: in Context with my main comment: please give the LOD converted to the atmospheric measurements (in µg/m^3)*

**Response:** The limit of detection of $NH_4^+$, $DMAH^+$, and $TMAH^+$ were 0.4, 4 and 2 ng $m^{-3}$ in ambient air, respectively.

*(10)    Line 150: is 0.002 μg/m^3 the LOD?*

**Response:** Yes. The data of 0.002 μg m$^{-3}$ was the LOD of TMA. It has been revised as "The TMA$_{gas}$ and TMAH$^+$ concentrations in PM$_{2.5}$ were mostly below the detection limit, varying at approximately 0.001±0.001 μg m$^{-3}$ (average ± standard deviation), regardless of the presence of offshore or onshore winds during short-term measurements in three seasons of 2019".

*(11)    Line 155: "extremely low" please be more precise here and give numbers. Also, please compare with other published amine measurements in marine regions (see main comment).*

**Response:** In the revision, it reads as "Figs 1a & b show that the TMA$_{gas}$ and TMAH$^+$ concentrations in PM$_{2.5}$ were mostly below the detection limit, varying at approximately 0.001±0.001 μg m$^{-3}$ (average ± standard deviation), regardless of the presence of offshore or onshore winds during short-term measurements in three seasons of 2019. The DMA$_{gas}$ and DMAH$^+$ concentrations varied at 0.018±0.021 and 0.017± 0.013 μg m$^{-3}$, respectively, which were approximately one order of magnitude larger than those of TMA$_{gas}$ and TMAH$^+$. This suggests that the TMA$_{gas}$ and TMAH$^+$ concentrations in the upwind continental and coastal atmospheres were substantially lower than those values over tens of ng m$^{-3}$ reported in the literature (Ge et al., 2011). Gibb et al. (1999) reported an even lower average of TMA$_{gas}$ (0.5 ng m$^{-3}$) and particulate TMAH$^+$ (0.5 ng m$^{-3}$) in the marine atmosphere over Arabian Sea on November 16 to December 19 in 1994. It is interesting that this was not the case – five to ten years ago in the atmosphere over the sea as listed in Table S1 and at the coastal sites (Yu et al., 2016; Xie et al., 2018)."

*(12)    Line 161: Explain the amines in relation to "onshore" and "offshore" winds more detailed and refer to the corresponding Figures.*

**Response:** To be clear, the sentence has been changed to **"**The DMA$_{gas}$ and DMAH$^+$ in PM$_{2.5}$ concentrations with offshore winds from the north were substantially higher than those with onshore winds from the south or southeast (the top of Fig 1a), suggesting that their continental emissions and related secondary sources were stronger." In the revision.

*(13)    Line 175: I am wondering if the times when the concentrations were below LOD are included in the given average values (e.g. for example setting the concentration to ½ LOD for times when the analytes could not be detected). Otherwise the mean data represent too high concentrations. A Table listing the mean (min-max) values for the gas and particulate amines for the different "campaigns" would help.*

**Response:** We thank the comments. In the revision, we consistently set the concentration to ½ LOD for times when the ions could not be clearly detected. In the origin version, the estimated values were used for those below LOD and had been corrected in the revision. The correction led to a change for a few lower concentration data and some averages. In addition, all Figures have been revised on this issue accordingly. In addition, the min-max values have been added in the revision.

*(14)        Line 185: From Fig.1 is looks as if DMA gas shows a similar behaviour as TMS gas. Please show the correlations reported in this passage (e.g in the supporting information). Otherwise, the differences between TMA gas and DMA gas are not easily understandable.*

**Response:** In the revised SI, the correlations between $TMA_{gas}$ and $DMA_{gas}$ were added in SI (Fig S2), in which three periods such as Peak 1, Peak 2 and Peak 3 were highlighted in different markers.

*(15)        Line 192: Please define "campaign A" and "campaign B" and the "costal station" more concise. Use the same descriptions concerning these separate "campaigns" consistently in the text and in the Figures. 20-22. Dec is "campaign B" = "port-anchoring period"?*

**Response:** Agree. Revised accordingly. Campaign A was conducted on 9-19 December, 2019. Port-anchoring occurred on December 19-22 when the vessel was anchored at the port while the sampling continued. Campaign B started from 27 December, 2019 to 17 January, 2020, organized by another research team.

*(16)        Line 227/228: Do these references state that the amines (or TMA) are transferred via primary sea spray? Please state this more clear. Connected to this: has a primary sea spray transfer (Line 236) been shown for (gaseous and/or particulate) amines?*

**Response:** The sentence has been revised as "This suggests that the observed $TMAH^+$ may not be derived from the neutralization reactions of TMA gas with acids in the marine atmosphere, and may have been derived from primary sea-spray organic aerosols (Hu et al., 2015, 2018). Primary sea-spray organic aerosols mainly contained primary and degraded organics (Ault et al., 2013; Prather et al., 2013; Quinn et al., 2015; Dall'Osto et al., 2019)."

*(17)        Line 256: It is highly speculative to comment on changing amine concentrations in the seawater, if such values were not measured . What means not "directly"? Same for line 265*

**Response:** Agree. The regression equation can allow to argue the emission potential, but it is highly speculated to argue the concentrations of $TMAH^+$ concentrations. In

the revision, the sentence has been revised as "However, the measured concentrations of TMAH$^+$ and seawater pH in the surface seawater were needed to confirm this".

*(18) Line 270: What is meant with "scenarios"? Do you mean "hypothesis"? Please explain.*

**Response:** Hypothesis is better and used in the revision.

*(19) Line 278: What is meant with "increasing" and "decreasing" period?*

**Response:** The sentence has been clarified as "The purple-red and dark-green markers represent the data obtained with increasing concentrations at 10:00 on 14 December - 22:59 on 16 December (increasing period) and with decreasing concentrations at 23:00 on 16 December -19:59 on 17 December (decreasing period) during Peak 3, respectively, which were analyzed separately."

*(20) Line 276 and following (chapter 4.2 and 4.3): I find it difficult to follow and understand the estimations and conclusions from the given information. I suggest adding some more details. Why is it justified to estimate the DMA$_{gas}$ in this way? This part is very descriptive and little explanatory. For example: The good correlation is mentioned (Line 280) but what can be concluded from that and why? What is the explanation that TMAH$^+$ decomposed into DMAH$^+$ (Line 283)? I have the feeling that the interrelationships and conclusion in 4.2 and 4.3 should be elaborated more strongly. The connections were much better illustrated in chapter 4.4.*

**Response:** Agree. The two parts indeed need to be improved. In this round, we make a substantial revision on Section 4.2 and 4.3 accordingly to clarify our approaches and findings.

*(21) Line 360: Did you exclude emissions of seabirds because the peaks were persistent for a long time under strong winds? Or what else is the reason? If so, maybe add ".. were therefore unlikely to be derived…" (Line 361)*

**Response:** Agree and revised accordingly.

*(22) Line 362: why "alternatively"?*

**Response:** The word has been removed.

*(23) Line 369: undetectable chemical conversion? What is meant by that?*

**Response:** It has been revised "Chemical conversion of particulate TMAH$^+$ by AIM-IC likely occurred and the products were undetectable by the AIM-IC. This requires further investigation."

---

## Author Response (AR2)

Dear Editor

Thank you very much. The revised manuscript has been polished using on-line language-editing service.

Your Sincerely.

Xiaohong

Prof. Xiaohong Yao (Ph.D)
Ocean University of China

*Thank you very much for the revision of your manuscript. Some more effort is needed before acceptance of the paper in APC. Please perform them and resubmit a revised manuscript for editor's review. kind regards the editor*
* * *
*In general, the manuscript could benefit from been read by a native English speaker. Herebelow, a number of areas where further rephrasing is needed.*
*Line 15: remove 'originating'*

**Response: Removed.**

*Gibb et al 1999 reference that has been added in the text is missing from the reference list.*

**Response: Added.**

*Text in lines 146-152 is not clear enough.*
*1) 'This suggests that the TMAgas and TMAH+ concentrations in the upwind continental and coastal atmospheres were substantially lower than those values over tens of ng m-3 reported in the literature (Ge et al., 2011). '*
*I suggest rephrasing it to:*
*$TMA_{gas}$ and $TMAH^+$ concentrations in the upwind continental and coastal atmospheres were substantially lower than the values reported in the literature of up to a few tens of ng.m$^{-3}$ (Ge et al., 2011). (is this what you want to say? )*

**Response: Revised.** The sentence has been revised as "$TMA_{gas}$ and $TMAH^+$ concentrations in the upwind continental and coastal atmospheres were substantially lower than those reported in the literature, by up to a few tens of ng m$^{-3}$ (Ge et al., 2011)."

*2) Gibb et al. (1999) reported an even lower average of TMAgas (0.5 ng m-3) and particulate TMAH+ (0.5 ng m-3) in the marine atmosphere over Arabian Sea on November 16 to December 19 in 1994.*

*Lower than what ?*

**Response:** The sentence has been revised as "However, Gibb et al. (1999) reported a low average of $TMA_{gas}$ (0.5 ng m$^{-3}$) and particulate $TMAH^+$ (0.5 ng m$^{-3}$) in the marine atmosphere over the Arabian Sea on November 16 to December 19, 1994."

*3) It is interesting that this was not the case–five to ten years ago in the atmosphere over the sea as listed in Table S1 and at the coastal 150 sites (Yu et al., 2016; Xie et al., 2018). For example, the concentrations of the two aminium ions were comparable in atmospheric particles collected at two other coastal sites located approximately 20 km from the study area (Xie et al., 2018).*
*These sentences need to be rephrased for clarity. What was not the case ? comparable to what ?*

**Response: Revised, it reads as** "Xie et al. (2018) reported that $TMAH^+$ concentrations were comparable to those of $DMAH^+$ in atmospheric particles collected at two other coastal sites located approximately 20 km from the study area, as listed in Table S1." The Table S1 have also been changed according to the sentence.

*Line 167: 'The comparison results strongly indicated that the $TMA_{gas}$ observed during Campaign A was largely derived from marine sources'*
*Please rephrase 'comparison results' and explain why they indicate that the $TMA_{gas}$ was mainly from marine sources?*

**Response: Revised, it reads as** "The observed $TMA_{gas}$ concentrations were one order of magnitude higher than those measured in the coastal atmosphere during the summer, fall, and winter. This suggested that the $TMA_{gas}$ observed during Campaign A was largely derived from marine sources rather than from long-range continental transport."

*Line 258-259: use 'from… to…' to define periods, for instance 'from 10. … to 23'*
*line 322: 'hourly average'*
*line 338: emissions from seabirds*

**Response: Revised.**